# Use of 18-Fluorodeoxyglucose Positron Emission Tomography and Near-Infrared Fluorescence-Guided Imaging Surgery in the Treatment of a Gastric Tumor in a Dog

**DOI:** 10.3390/ani14202917

**Published:** 2024-10-10

**Authors:** Su-Hyeon Kim, Yeon Chae, Byeong-Teck Kang, Sungin Lee

**Affiliations:** 1Department of Veterinary Surgery, College of Veterinary Medicine, Chungbuk National University, Cheongju 28644, Republic of Korea; ssuekim57646@gmail.com; 2Department of Veterinary Surgery, Heamaru Referral Hospital, Seongnam 13590, Republic of Korea; 3Laboratory of Veterinary Internal Medicine, College of Veterinary Medicine, Chungbuk National University, Cheongju 28644, Republic of Korea; bluesfiddle@naver.com (Y.C.); kangbt@cbu.ac.kr (B.-T.K.)

**Keywords:** fluorodeoxyglucose, positron emission tomography/computed tomography, indocyanine green, near-infrared, gastric tumor, dog, canine, leiomyosarcoma

## Abstract

**Simple Summary:**

Gastric tumors in dogs, although uncommon, present considerable diagnostic and treatment challenges, often diagnosed late in the disease course, resulting in poor prognosis. Near-infrared (NIR) fluorescence imaging with indocyanine green (ICG) and positron emission tomography (PET) using 18F-fluorodeoxyglucose (FDG) offer promising approaches in human medicine for intraoperative tumor detection and metastasis assessment. This case report presents a novel application of ICG-guided surgery and 18F-FDG PET/CT in a canine gastric tumor.

**Abstract:**

A 13-year-old Maltese dog with an abdominal mass underwent 18F-FDG PET/computed tomography (CT) for tumor localization and metastatic evaluation. PET/CT scans revealed a gastric mass near the esophagogastric junction and demonstrated mean and maximum standardized uptake values (SUVs) of 4.596 and 6.234, respectively, for the abdominal mass. Subsequent surgery incorporated ICG for NIR fluorescence-guided imaging, aiding in precise tumor localization and margin assessment. The excised mass was identified as a low-grade leiomyosarcoma on histopathology. The dog underwent PET/CT imaging six months postoperatively following the excision of the mass, which confirmed the absence of recurrence or residual lesions during follow-up. NIR fluorescence imaging using ICG demonstrated efficacy in real-time tumor visualization and margin assessment, a technique not previously reported in veterinary literature. The PET/CT findings complemented the diagnosis and provided valuable insights into metastasis. The absence of recurrence or complications in postoperative follow-up underscores the potential of these imaging modalities in enhancing surgical precision and improving prognosis in canine gastric tumors.

## 1. Introduction

Gastric tumors are uncommon in dogs, comprising <1% of all canine neoplasms [1]. In dogs, malignant tumors such as adenocarcinomas, leiomyosarcomas, and lymphomas have been reported more than benign tumors, with adenocarcinomas representing the majority of cases [2]. The diagnosis is difficult and often occurs late in the course of the disease, which contributes to the poor prognosis because of metastasis [3]. Furthermore, although complete surgical resection is the main treatment for gastric cancer [4], tumors located in anatomically challenging regions, such as near the esophagus and those that metastasize to regional lymph nodes, often pose challenges to the completion of surgical resection.

Positron emission tomography (PET) using 18F-fluorodeoxyglucose (FDG) is a tool that provides the status of tumor metastasis and cell biology [5]. 18F-FDG, a glucose analog, is the most used radiotracer for the detection of malignancies in PET/computed tomography (CT) [6]. Compared to CT alone, PET-CT gives more precise data regarding metabolic information and tumor staging [7]. In human medicine, PET/CT has also been utilized for the detection and confirmation of gastric tumor recurrence after curative resection [5,7].

Near-infrared (NIR) fluorescence imaging is a promising image-guided surgical method used to facilitate the intraoperative detection and evaluation of surgical margins. NIR light (700–800 nm) can provide tumor-specific fluorescent contrast because of its high tissue penetration and low autofluorescence in biological tissues. Indocyanine green (ICG) is the first Food and Drug Administration-approved fluorescent agent that responds to NIR [8], exhibiting an absorption peak of approximately 807 nm and an emission peak of approximately 822 nm in plasma, both of which are within the range of the NIR window [9]. Because of its safety, relatively low cost, and well-established clinical applications, ICG has been used as a fluorescent marker in general surgery clinical settings, such as in identifying tumors and surgical margins in human medicine [10,11].

To the best of our knowledge, there are no previous reports in veterinary literature regarding the use of 18F-FDG PET/CT for identifying gastric tumors in dogs. Additionally, while NIR fluorescence imaging with ICG has been explored in human medicine for various applications, its specific use for gastric tumors in veterinary practice has not been documented. This case report aims to highlight the dual objectives of utilizing 18F-FDG PET/CT for tumor detection and metastasis evaluation during the diagnostic phase and ICG-guided NIR fluorescence imaging during therapy to achieve precise tumor margin identification during gastrectomy. Additionally, 18F-FDG PET/CT was employed in the follow-up phase to confirm the absence of tumor recurrence.

## 2. Case Presentation

A 13-year-old castrated male Maltese dog weighing 3.5 kg was referred for an abdominal mass, which was incidentally identified via ultrasonography when the referring veterinarian was examining the dog for suspected bladder stones. The dog displayed inappetence, with normal heart rate, respiratory rate, Doppler-induced blood pressure, and rectal temperature. A history of chronic vomiting was not reported. A complete blood count, serum biochemical, and electrolyte analyses showed normal values for all parameters except the elevation of alkaline phosphate levels (176 U/I; reference range, 0–97 U/I).

Abdominal ultrasonography revealed an anechoic mass, likely in the lesser curvature of the stomach, with no vascularization on color Doppler examination and multiple cystoliths. Ventrodorsal, right lateral, and left lateral thoracic radiographs were unremarkable, except for multiple large calculi in the urinary bladder. 18F-FDG PET/CT (PET/CT; Discovery-72 STE, General Electric Medical Systems, Waukesha, WI, USA) was scheduled to identify the exact tumor location and metastatic state.

The dog underwent a fasting period of at least 12 h before the scan. Blood glucose levels assessed just before the PET scan were within the normal range (105 mg/dL; reference range 81–133 mg/dL). The dog received premedication with midazolam (0.2 mg/kg; Midazolam, Bukwang Pharm. Pharm. Co., Ltd., Seoul, Republic of Korea) following intravenous catheterization and placement of an indwelling urinary catheter. Anesthesia was induced using intravenous propofol (6 mg/kg; Provive, Myungmoon Pharm. Co., Ltd., Seoul, Republic of Korea), followed by endotracheal intubation. The dog was maintained under general anesthesia with isoflurane (Terrell, Piramal Critical Care, Bethlehem, PA, USA) and ventilated with 100% oxygen. Following anesthesia induction, the dog was positioned in sternal recumbency and maintained in this position throughout both CT and PET scans. CT images, both pre- and post-contrast, were obtained before the PET scan using a Muti-detector CT scanner (100 mAs, 120 kVp, 1.25 mm slice thickness). An approximately 2 cm long × 1.5 cm wide oval-shaped mass was localized at the cardiac region of the lesser curvature of the stomach, which was on the esophagogastric junction. Contrast scans were performed via intravenous administration of 880 mL/kg iohexol (Omnipaque; GE Healthcare, Marlborough, MA, USA). Post-contrast CT showed slightly heterogeneous contrast enhancement. Considering that a contrast-enhanced mucosal layer was observed surrounding the mass after contrast, the mass was suspected to be in the submucosa or the muscular layer with an intact mucosal layer. The PET/CT scan was conducted following the intravenous administration of 0.59 mCi (0.17 mCi/kg) 18 F-FDG, with a subsequent 45 min uptake period. Emission scans were acquired as static frames for approximately 2 min per bed position, totaling five bed positions. The CT and PET scans were acquired using a single device. There was an average time lapse of approximately 1 h between obtaining CT and PET images. The overall anesthesia duration for both CT and PET imaging was approximately 3 h. The analysis of PET images was carried out using a commercial program (OsiriX MD v10.0; Pixmeo Sarl, Geneva, Switzerland). The region of the suspected mass showed a high level of 18F-FDG uptake (Figure 1A). The regions of interest were drawn manually on the PET/CT fusion images. The standardized uptake value (SUV) is a measurement that normalizes tissue 18F-FDG uptake (MBq/mL) relative to the injected dose (MBq) per body weight (g). The calculated maximum and mean SUVs of the abdominal mass were 6.234 and 4.596, respectively. There was no evidence of metastatic lesions throughout the body, including the liver, spleen, lungs, lymph nodes, and regional lymph nodes, based on the whole-body PET/CT images.

Surgical resection was recommended, and partial gastrectomy via a midline abdominal approach was planned. ICG (25 mg, Cellbiongreen, Cellbion Co., Ltd., Seoul, Republic of Korea) was prepared by dissolving it in 5 mL of sterile water, resulting in a 5 mg/mL concentration of ICG. Subsequently, 1 mL of the dissolved solution was combined with an additional 9 mL of saline water, resulting in a final concentration of 0.5 mg/mL of ICG. The solution was prepared to inject into the submucosal layer at four sites (proximal, distal, and bilateral to the tumor region) surrounding the mass during surgery. An intraoperative NIR fluorescence-guided imaging system (ZNI; Metaple Bio, Seoul, Republic of Korea) was used for ICG display.

The dog was positioned in dorsal recumbency, and a celiotomy was performed from the xiphoid to the pubis, along with a left paracostal incision by extending through the external abdominal oblique muscle to access the cardiac region of the stomach. After palpating the mass at the esophagogastric junction, stay sutures were applied, and a gastrotomy was performed to access the tumor located in the mucosal layer of the stomach. The stomach mass was identified on the esophagus–gastric junction (Figure 2). Submucosal local injections of ICG were performed around the quadrants of the stomach mass using a 1 mL syringe with a 26-gauge needle at a concentration of 0.5 mg/mL and a total volume of 1 mL (Figure 3A and Figure 4A). To prevent leakage after the injection, the site was immediately compressed with surgical gauze. Within 1 min of ICG application, greenish discoloration was observed near the esophagus–cardia junction, in addition to the previous mass location (Figure 4B). Before resecting the mass, the location of the ICG-injected region (Figure 4C) was observed using a NIR fluorescence camera (Figure 4D). Thirty minutes after the ICG injection, we made an incision and removed the mass, which was identified intraoperatively using NIR fluorescence imaging (Figure 3B,C). Warm sterile saline was locally injected submucosally around the stomach mass to elevate it from deeper layers. The mass was then isolated by blunt dissection using an electric cautery (Monopolar; Covidien, Mansfield, MA, USA) and a bipolar vessel sealing device (LigaSure; Covidien, Mansfield, MA, USA). The excised mass was immediately observed under the NIR fluorescence imaging camera ex vivo to confirm the presence of ICG fluorescence (Figure 5). The stomach wall was closed in two layers using simple continuous sutures and simple interrupted sutures with 3/0 absorbable monofilament (PDS II 3-0; Ethicon, Raritan, NJ, USA). The abdominal wall was closed with 3/0 absorbable monofilament (PDS II 3-0; Ethicon, USA), the subcutaneous tissue with 4/0 absorbable monofilament (PDS II 4-0; Ethicon, USA), and the skin was closed using a skin stapler (Covidien Skin Stapler; Medtronic, USA). The dog recovered successfully from the anesthesia.

The surgically resected mass was preserved in 10% neutral buffered formalin for 24 h before being shipped to the laboratory (IDEXX Laboratories, Inc., USA) for histopathological examination. The sample was processed, embedded in paraffin, sectioned at 4 µm thickness, and stained with hematoxylin and eosin (H&E) for routine histological assessment. Pathological evaluation revealed the characteristics of mesenchymal gastrointestinal neoplasia. Further immunohistochemical analyses were performed using CD117 (c-Kit, clone EP10), DOG-1 (clone SP31), and SMA (clone ASM-1) for a definitive diagnosis. The results of the histopathological and immunohistochemical examinations were most compatible with low-grade leiomyosarcoma (Figure 6). The neoplastic cells were immune-negative for CD117 (c-Kit) and DOG1 but displayed strong cytoplasmic immunolabeling for smooth muscle actin (aSMA), confirming the diagnosis. The mitotic count was 0 in 2.37 mm^2^, and the tumor was completely excised with clean margins. No vascular or lymphatic invasion was observed.

The possibility of adjuvant targeted therapy with toceranib and imatinib was considered; however, it was not pursued due to the absence of CD117 expression in the tumor cells. Additionally, the low-grade leiomyosarcoma was completely excised, with no signs of metastasis or lymphatic or vascular invasion. Thoracic radiography and abdominal ultrasonography were performed every 3 months for regular postoperative follow-up. The dog underwent follow-up PET/CT scans 6 months after surgery. The blood glucose level was checked before the PET/CT examination, and normal blood glucose (116 mg/dL; reference range 81–133 mg/dL) was confirmed. Notably, increased FDG uptake previously observed in the gastric mass was not detected, indicating any recurrence or residual lesions (Figure 1B). At the time of writing this paper, 2 years after the surgery, the dog has not shown any further complications or evidence of metastasis or local recurrence.

## 3. Discussion

This study highlights the use of 18F-FDG PET/CT for tumor localization and metastasis evaluation alongside ICG-guided NIR fluorescence imaging for precise tumor margin identification in a case of canine gastric leiomyosarcoma. While these techniques show promise, further research is necessary to fully evaluate their effectiveness in detecting metastases and recurrence in veterinary oncology.

In human oncology, 18F-FDG PET has been used in the following areas: staging tumors [6], confirming metastases [12], and confirming recurrence following surgery or radiotherapy [13]. Although 18F-FDG findings of gastric tumors have been reported in human medicine, to the best of our knowledge, 18F-FDG PET characteristics in canine gastric tumors have not been reported in veterinary medicine. In a study on the physiological uptake of 18F-FDG in normal dogs, the maximum and mean SUVs were 1.70 ± 0.36 and 0.89 ± 0.25, respectively [14]. In this case, the mean and maximum SUVs were 4.596 and 6.234, respectively, which are substantially higher than those of normal dogs. Because data on PET/CT in veterinary medicine are limited, there was only one case of the gastric tumor identified by 18F-FDG for comparison [15]. Therefore, direct comparisons with SUVs from different institutions using the provided data should be approached cautiously. Further research is necessary to quantify SUVs specific to canine gastric tumors.

Surgical resection is regarded as a crucial and curative therapy for gastric tumors; chemoradiation, perioperative and adjuvant chemotherapy, and expanded lymph node dissection can all improve the prognosis of stomach cancer [1]. In human medicine, FDG-PET/CT encounters challenges in detecting primary gastric cancer, especially in early stages and with specific histological types, with physiological uptake and potential masking by conditions such as gastritis [16]. For staging gastric cancer, FDG-PET/CT has restricted utility. Although FDG-PET/CT may possess enhanced specificity for detecting loco–regional lymph nodes compared to CT alone, its sensitivity is relatively lower [6,16]. However, in restaging after surgical resection, FDG-PET/CT exhibits sensitivity and specificity of 85% and 78%, respectively, with high detection rates for recurrent cases [6,16]. Despite its limitations in primary tumor detection and lymph node staging, FDG-PET/CT excels in the recurrence detection of gastric tumors, highlighting its potential application for recurrence detection in veterinary medicine. The extended duration of anesthesia remains a concern, particularly for critical patients with underlying conditions such as renal or cardiovascular disease [17]. This poses a limitation for broader application in such cases.

In human medicine, the principal surgical method for curative surgery is standard gastrectomy, which requires resecting at least two-thirds of the stomach with lymph node dissection [18]. However, the surgical strategy of standard gastrectomy is controversial owing to its poor prognosis [19]. Furthermore, non-standard gastrectomy can also be considered, in which different degrees of stomach resection and/or lymphadenectomy are performed with curative intent [18]. Recent studies reported that proximal gastrectomy has survival rates similar to those of total gastrectomy, with the added advantage of maintaining the gastric remnant [20]. The extent of the surgical margin is significant for nutritional outcomes and quality of life in the later postoperative stage, as well as morbidity and mortality in the early postoperative phase [21]. To enhance the chances of organ preservation after gastrectomy, precise tumor localization is crucial. In other words, if the tumor location and surgical margin are correctly diagnosed, partial gastrectomy can be performed rather than total gastrectomy.

Several methods have been developed for precise tumor detection and minimally invasive surgery in the past few years. Recent research has demonstrated the potential of fluorescence-based imaging-guided surgery to detect tumors intraoperatively [22]. In veterinary medicine, surgery is the most invasive but also the most effective method of gastric neoplasia treatment [1]. Therefore, securing the appropriate surgical margin was important for an improved prognosis of gastric neoplasia in the dog in the present case.

Fluorescence-guided surgery (FGS) using ICG is an innovative technique for the real-time visualization of tumors and resection margins [9]. ICG, which was used in the present study, is a suitable fluorophore for tumor tattooing, as it has minimal side effects, an extended absorption duration, and the potential for enhanced detection via NIR fluorescence [9]. Visualization of precise surgical margins would reduce the risk of recurrence and eliminate unnecessary tissue loss, allowing a better prognosis [23]. In human medicine, intraoperative ICG FGS has been extensively investigated and expanded across various clinical uses over the past few years [24]. ICG has been used to detect hepatocellular carcinoma and colon cancer metastases, suggesting image-guided resection margins with an image of the NIR fluorescence signal [25]. Identification of breast tumors by NIR fluorescent surgery using ICG for breast conservation has also been reported [26]. According to the study protocol, a diluted solution of ICG was injected into the margin of the breast tumor at 4–8 different locations. Remarkably, the ICG selectively accumulated in the tumor tissue while sparing the surrounding healthy tissue. This property of ICG has also been exploited in the detection of gastric cancer and lymph node metastasis [27]. In veterinary medicine, however, the number of clinical trials utilizing ICG-guided surgery for tumor resection, tumor identification, and evaluation of surgical margins for complete resection are limited compared with those in humans. One report documented a case involving the resection of a cutaneous mast cell tumor in a dog using NIR fluorescence imaging with ICG, highlighting the potential application of this technique for complete resection of the tumor [28]. Ida et al. demonstrated the feasibility of NIR fluorescence imaging for intraoperative mapping of hepatocellular carcinomas in dogs, potentially improving the chances of achieving complete resection with hepatic tumors [29].

Recently, NIR FGS with ICG has been applied to assess the tumor location to achieve negative surgical margins during gastric surgery [30]. The injection time of ICG varies from (i) the day before surgery to (ii) minutes before dissection, as seen in our case [30,31,32]. In human medicine, a preoperative submucosal injection of ICG is performed in a minimally invasive manner under endoscopic guidance. However, the dog in this case report, weighing approximately 3 kg, was not indicated for endoscopy because of its small size, and open surgery was recommended. Therefore, as a minimally invasive approach for preoperative injection was not possible, ICG was administered approximately 30 min before resection. The dosage of ICG was based on a previous study [27], which demonstrated its efficacy for fluorescence-guided surgery. In total, 1 mL of a 0.05 mg/mL ICG solution was injected submucosally into four quadrants surrounding the tumor with caution of leakage. The two-step dilution protocol was utilized to ensure accurate dosing and minimize potential errors [33]. Under an infrared fluorescence camera, the gastric mass, including parts of the cardia and gastroesophageal junction, exhibited ICG fluorescence, and the surgical margin was determined based on the ICG fluorescence. However, it is crucial that the ICG injection technique is performed with utmost precision to avoid penetrating the tumor, as this could cause bleeding, obscure the surgical field, and potentially spread cancer cells. While we did not observe complications in this case, it is important to acknowledge that blurring of the surgical site by ICG is a reported complication that may complicate surgery [17]. However, compared to traditional agents like methylene blue, the risk of such complications may be lower with ICG, even in instances where a puncture occurs outside the digestive tract, as there is no observable color change once the ICG spreads to the peritoneal cavity [27]. This aspect proves advantageous, as it does not disrupt the surgical site visually, which makes it suitable for implementation in our case. Nevertheless, there is still considerable clinical and technical variability when using ICG, such as dosage, concentration, and timing of administration. Therefore, further investigations are required to determine the optimal injection timing and dosage of ICG.

## 4. Conclusions

To the best of our knowledge, this is the first case report utilizing 18F-FDG PET/CT in a canine gastric tumor. While this study presents the potential of PET/CT for tumor detection and staging, the role of this technique in detecting recurrences requires further investigation in a larger cohort. Moreover, this case highlights the novel application of ICG as a NIR fluorescence agent for intraoperative identification of gastric tumors and precise surgical margin assessment during gastrectomy. The use of NIR fluorescence imaging with ICG demonstrates the potential to enhance tumor margin evaluation, but additional studies are necessary to validate its effectiveness in veterinary oncology. Overall, this case provides preliminary insights into the potential roles of both 18F-FDG PET/CT and ICG-guided NIR fluorescence imaging in canine gastric tumor management.

## Figures and Tables

**Figure 1 animals-14-02917-f001:**
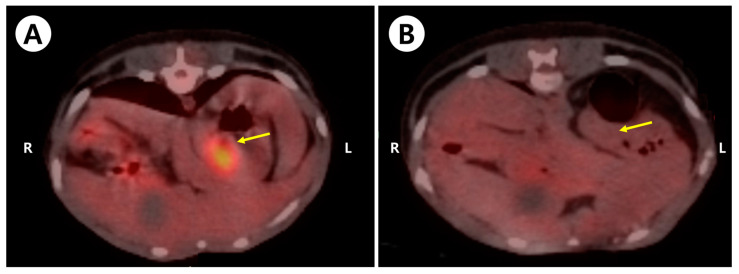
Representative 18F-2-deoxy-2-fluoro-D-glucose (FDG) positron emission tomography (PET)/computed tomography (CT) images of a dog with a gastric tumor. (**A**) Increased 18F-fluorodeoxyglucose uptake in the gastric mass (yellow arrow) is observed in the transverse plane before surgery; (**B**) after surgery, the increased FDG uptake previously observed in the gastric mass is not observed (yellow arrow).

**Figure 2 animals-14-02917-f002:**
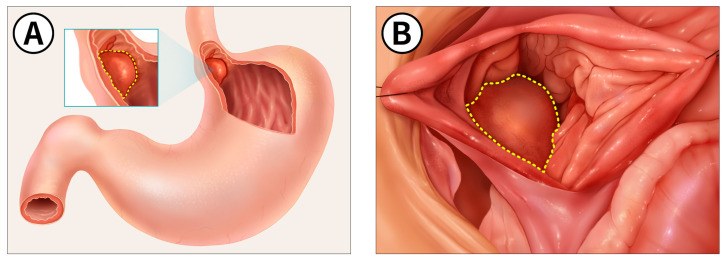
(**A**) Schematic representation of the gastric tumor that was located on the esophagogastric junction; (**B**) visualization of the gastric tumor after a gastrotomy incision. The yellow dotted line represents the part of the mass that was initially detected visually.

**Figure 3 animals-14-02917-f003:**
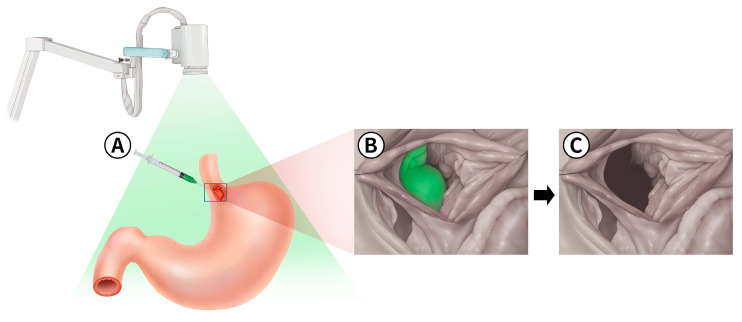
(**A**) Administration of indocyanine green (ICG) into the tumor of the esophagogastric junction, located at the cardiac region of the lesser curvature of the stomach; (**B**) intraoperative near-infrared (NIR) fluorescence imaging, revealing the fluorescence emitted by ICG within the gastric tumor. The fluorescence aids in precise tumor localization during the surgical procedure; (**C**) post-resection image of the gastric tumor, demonstrating the absence of ICG fluorescence.

**Figure 4 animals-14-02917-f004:**
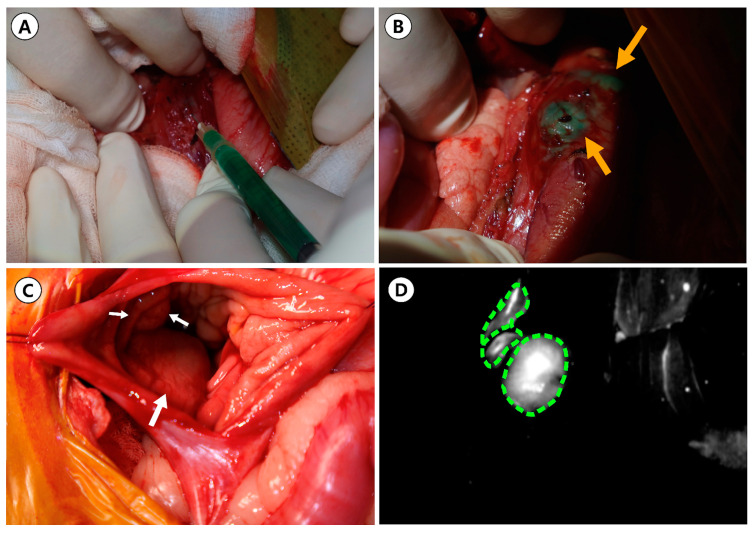
(**A**) Indocyanine green (ICG) was injected into the gastric submucosa in the quadrants surrounding the gastric tumor at a dosage of 1 mL (0.5 mg/kg); (**B**) intraoperative mass detection using ICG was performed, revealing a green mass (orange arrow) located at the cardiac region of the lesser curvature of the stomach, extending toward the esophagogastric junction. The tumor location was visualized using near-infrared light-activated ICG fluorescence: (**C**) under white light and (**D**) under near-infrared light-activated ICG. White arrows indicate the gastric mass within the stomach wall. The green dotted line represents the fluorescence emitted by the gastric tumor.

**Figure 5 animals-14-02917-f005:**
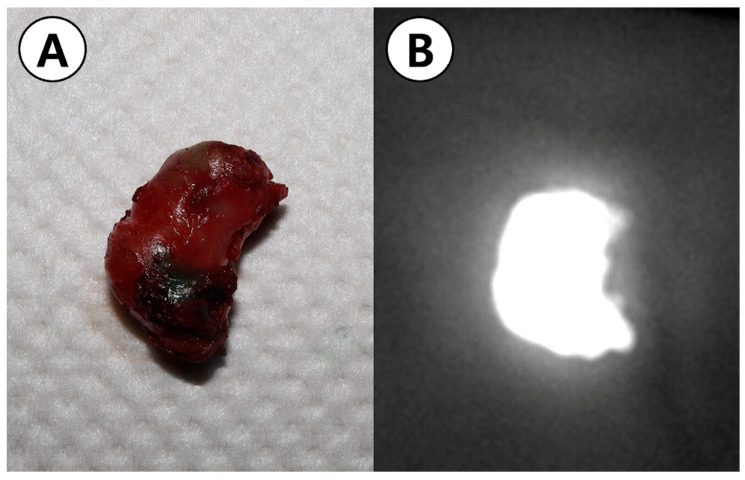
(**A**) Image of the resected stomach mass; (**B**) representative brightfield image of the resected stomach mass under near-infrared light-activated indocyanine green.

**Figure 6 animals-14-02917-f006:**
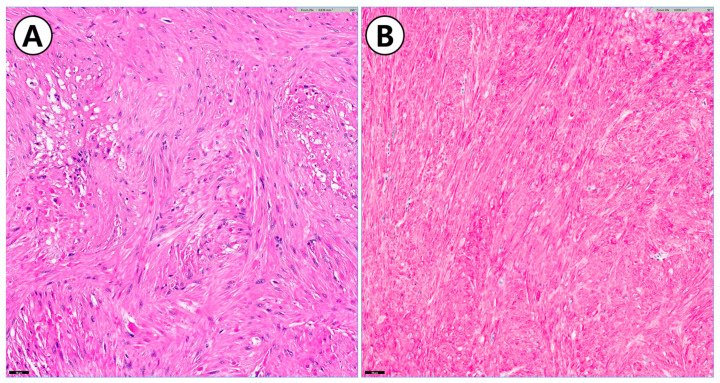
Histopathological and immunohistochemical evaluation of a gastric mass. (**A**) Mesenchymal gastrointestinal neoplasia showing neoplastic spindle cells arranged in interlacing bundles and streams on a dense stroma (20×); (**B**) immunohistochemical staining for aSMA showing positive, moderate to strong intracytoplasmic highlighting of neoplastic spindle cells, consistent with leiomyosarcoma (20×).

## Data Availability

The original contributions presented in the study are included in the article. Further inquiries can be directed to the corresponding author.

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
