# Peer review of "Use of 18-Fluorodeoxyglucose Positron Emission Tomography and Near-Infrared Fluorescence-Guided Imaging Surgery in the Treatment of a Gastric Tumor in a Dog"

_animals, 2024, doi:10.3390/ani14202917_

Round 1
Reviewer 1 Report
Comments and Suggestions for Authors
The topic of the article is very interesting, and the case report may enrich the clinical knowledge regarding the imaging and surgical management of neoplastic lesions in veterinary medicine. However, in my opinion, the article needs major revisions. My major concerns relate to the organization of the case description and especially to the definition of the aims related to the two techniques used perioperatively and intraoperatively. Throughout the manuscript - in the introduction, results, and especially in the discussion - the advantages and disadvantages of 18F-FDG PET/CT (for preoperative diagnosis and follow-up evaluation) and the intraoperative use of ICG as an NIR fluorescence agent (for safe and complete mass excision) should be presented more schematically. Please, check the references as there are mismatches with the citations mentioned in the text.

Author Response
Title
Comment 1: The title of the article may be unclear. I suggest the following change to better reflect the dual aim of the case description: Use of 18-Fluorodeoxyglucose positron emission tomography and near-infrared fluorescence-guided imaging surgery in the treatment of a gastric tumor in a dog.
Response 1: We thank the reviewer for this suggestion. The title was revised as suggested (page 1, line 2-4 in the revised manuscript).
Abstract
Comment 2: Lines 25-26: please, rephrase the sentence. The current sentence implies that the neoplasia was imaged, but I believe it should indicate that the dog was imaged during post-operative follow-up.
Response 2:
We sincerely thank the reviewer for the insightful suggestion, which has contributed to enhancing the clarity of our manuscript. We recognize that the original wording may have caused some confusion by implying that the neoplasia was imaged, rather than the dog during post-operative follow-up. Following your recommendation, we have rephrased the sentence (page 1, lines 24-26 in the revised manuscript).
Introduction
Comment 3: Lines 46-55: I suggest swapping this section with lines 56-62, so that the PET technique is described before the NIR imaging. Please maintain the same in the case description, results, discussion, and conclusions.
Response 3: We appreciate the reviewer’s valuable suggestion. We have revised the manuscript accordingly, swapping the sections as recommended so that the PET/CT technique is described before the NIR imaging (page 2, lines 45-61).
Comment 4: Line 55: Please, specify here and throughout the text whether you are referring to human medicine or veterinary medicine. Moreover, one or more references are needed.
Response 4: We thank the reviewer for this valuable suggestion. To address the concern, we have clarified the text throughout, as requested, and have added the appropriate reference (page 2, line 61 in the revised manuscript) to further support the information provided.
Comment 5: Line 64: did you state that the 18-FDG PET/CT technique was not described for gastric tumors? What about the NIR? Please specify the described use in terms of human versus veterinary medicine and distinguish between different clinical applications (e.g., oncology versus non-oncological uses, gastric tumors versus other neoplasms, primary tumors versus lymph node involvement).
Response 5: We appreciate the reviewer’s insightful comment. The sentence has been revised for clarity (page 2, lines 62-66 in the revised manuscript).
Comment 6: Lines 63-71: Please provide a clearer exposition of the study’s dual aim. The study employs two techniques at different phases, for diagnosis, therapy, and follow-up, each serving distinct purposes.
Response 6: We appreciate the reviewer’s valuable feedback. We have clarified the dual aims of the study (page 2, lines 62-70 in the revised manuscript).
Case presentation
Comment 7: Line 74: How was the abdominal mass diagnosed by the referring veterinarian? Were the ultrasound and radiological examinations (lines 80-83) performed by the referring vet?
Response 7: We thank the reviewer for the suggestion. We have clarified how the abdominal mass was initially identified and revised the sentence to reflect this (page 3, line 76-78 in the revised manuscript) as follows:
Comment 8: Line 84: please, replace “performed” with “scheduled”.
Response 8: We thank the reviewer for the suggestion. The sentence has been revised as requested, replacing "performed" with "scheduled" (page 3, lines 87-88 in the revised manuscript).
Comment 9: Line 90-91: please, describe the anesthetic procedure (induction, maintenance).
Response 9: We appreciate the reviewer’s suggestion. In response, we have provided additional details regarding the anesthetic procedure (page 3, lines 93-97 in the revised manuscript).
Comment 10: Line 100: could you clarify what is meant by the “submucosa of the muscular layer”? Did the mass spread into the submucosa, or did it extend deeper into the muscle layer?
Response 10: We appreciate the reviewer’s helpful comments. To clarify, the mass was suspected to be located either within the submucosal layer or extending into the muscular layer, with the mucosal layer remaining intact. We have revised the sentence accordingly (page 3, line 106 in the revised manuscript).
Comment 11: Lines 123-127: why did you use this two-step dilution protocol instead of directly diluting to obtain the desired concentration? Please, explain this choice in the section of the discussion related to the doses of ICG.
Response 11: We appreciate the reviewer’s comment which could enhance the clarity of our manuscript. This approach was chosen to ensure accurate dosing. By preparing a concentrated solution first and then diluting it further, we could achieve greater precision in the final concentration of ICG. We have added an explanation of this choice to the discussion section related to the dosing of ICG (page 8, lines 267-270 in the revised manuscript).
Comment 12: Line 132: please, specify if you performed a cranial celiotomy.
Response 12: We thank the reviewer for insightful suggestion. To enhance the accuracy of description regarding the surgical approach, we have revised this sentence (page 4, lines 139-143 in the revised manuscript).
Comment 13: Line 133: based on Figure 2, it appears that you performed a gastrotomy before identifying the mass. Could you clarify if this was the case?
Response 13: We appreciate your valuable feedback. To clarify, we have added a sentence to enhance clarity regarding the surgical procedure (page 4, lines 142-143 in the revised manuscript).
Comment 14: Line 135: please include details about the instruments used for the injection, specifically the size of the needle.
Response 14: We thank the reviewer for the helpful suggestion which has contributed to the clarity of our methods. We have added details regarding the instruments used for the injections (page 4, lines 145-148 in the revised manuscript).
Comment 15: Line 138: please, replace gastrostomy with gastrotomy.
Response 15: Thank you for your comment. We revised this sentence as requested (page 4, line 150 in the revised manuscript).
Comment 16: Line 147: please, specify the suture material and pattern used.
Response 16: We appreciate the reviewer’s suggestion. The suture material and pattern have been specified in the revised manuscript (page 4, lines 159-161 in the revised manuscript).
Comment 17: Line 170-176: I recommend providing a more detailed description of the histological section. Specifically, include information on how the sample was preserved (e.g., 10% formalin) and for how long before examination, and how it was shipped to the laboratory. Additionally, specify which histological stains were used, what aspects were assessed, and the corresponding results.
Response 17: Thank you for your insightful comment. We have revised the manuscript to include a more detailed description of the histological section as suggested (page 7, lines 184-195 in the revised manuscript).
Comment 18: Lines 186-187: please, replace with “6 months after surgery”
Response 18: We appreciate the reviewer’s suggestion. We revised the sentence as requested (page 7, lines 206-207 in the revised manuscript).
Discussion
Comment 19: I recommend starting the discussion with a concise sentence highlighting the most significant results, particularly in relation to the dual aim of the study.
Response 19: We appreciate the reviewer’s valuable suggestion. In the revised manuscript, we have incorporated a concise statement at the beginning of the discussion to emphasize the study’s most significant findings, particularly in relation to its dual aims (page 7, lines 214-218 in the revised manuscript).
Comment 20: Line 201: please, specify the meaning of the acronym SUV before using it for the first time.
Response 20: We appreciate the reviewer’s observation. However, we have already defined the acronym "SUV" as "standardized uptake value" in lines 116–117 of the revised manuscript before its first use in line 224. Therefore, no additional changes were made in this regard.
Comment 21: Line 203: please, add the reference of the other case of gastric tumor identified with 18F- FDG.
Response 21: We apologize for the oversight and appreciate the reviewer’s feedback. The reference has now been added as requested (page 8, line 228 in the revised manuscript).
Comment 22: Line 218: It has been reported that longer durations of anesthesia are associated with an increased risk of death in dogs. You mentioned that the animal was under anesthesia for three hours during the diagnostic procedures, and I believe this could be an important limitation of using this technique, especially in critical patients. Please discuss this aspect.
Response 22: We appreciate the reviewer’s insightful comment. As a result of refining our protocol, we have since reduced the anesthesia duration to approximately two hours by administering the FDG injection prior to anesthesia. However, we acknowledge that there are limitations when using this technique in patients with conditions such as renal failure or heart disease, which pose higher risks during anesthesia. We have revised the manuscript that incorporate this limitation into our discussion (page 8, lines 243-246 in the revised manuscript).
Comment 23: Lines 239-241: please, provide a reference for this sentence.
Response 23: Thank you for the suggestion. We have located an appropriate reference to support this (page 8, line 271 in the revised manuscript).
Comment 24: Line 267: please, delete the full stop (.) after the citation [22].
Response 24: We appreciate the reviewer’s attention to detail. We revised the sentence as requested (page 9, lines 297-298 in the revised manuscript).
Comment 25: Line 269: in my opinion, it should be discussed that the ICG injection technique must be performed with the utmost precision to avoid penetrating the tumor, which could cause bleeding, obscure the surgical field, and potentially spread cancer cells.
Response 25: Thank you for your valuable feedback. We completely agree that the ICG injection technique requires careful precision to prevent any complications, including tumor penetration, which could lead to bleeding, obscure the surgical field, or potentially facilitate the spread of cancer cells. We have revised the manuscript to emphasize the importance of meticulous technique during ICG administration, as suggested (page 9, lines 303-306 in the revised manuscript).
Comment 26: Line 271: replace “can be determined” with “were determined”.
Response 26: We appreciate the reviewer’s suggestion. We revised the sentence as requested (page 9, line 303 in the revised manuscript).
Comment 27: Lines 272-274: please, move this sentence at the beginning of the section where you describe the FGS ICG technique (line 239). Moreover, explain the mechanism by which ICG accumulates in tumor tissues.
Response 27: We appreciate your valuable feedback. In response to your suggestion, we have revised the manuscript (page 8, lines 267-270 in the revised manuscript).
Comment 28: Lines 274-276: how can you affirm this if such complication did not occur in the present case? Moreover, blurring of the surgical site by ICG is a reported complication which can make the surgery more complex (Beer et al., 2022).
Response 28: Thank you for your insightful comments. We acknowledge that the lack of complications in the present case limits our ability to make definitive assertions regarding ICG's safety. In response, we have revised the text to clarify that our comparison to methylene blue and similar agents suggests a potentially reduced likelihood of complications when using ICG. We have also addressed the concern regarding blurring of the surgical site, as noted in Beer et al. (2022) (page 9, lines 303-311 in the revised manuscript).
Conclusion
Comment 29: In my opinion, it cannot be stated that 18F-FDG PET plays an important role in detecting tumor recurrences, as there were no recurrences in this case and no o statistical comparison was made with the sensitivity of other diagnostic techniques more commonly used in veterinary medicine. Rather, I would suggest that this technique needs to be studied in a larger number of patients, to determine if it can aid in detecting metastases, as it does in human medicine. Additionally, I recommend outlining the conclusions in relation to the dual aim of the study. I would rephrase the conclusions keeping this in mind.
Response 29: We appreciate your insightful comments. We have revised the manuscript to address the dual aims of the study as suggested (page 9, lines 317-327 in the revised manuscript).
Figure 1
Comment 30: In Please, reverse the image by inverting the right and left sides. If possible, provide sharper and brighter pictures.
Response 30: Thank you for the suggestion. We have reversed the image as recommended, with the right and left sides inverted. However, we sincerely apologize as the current image resolution is the best achievable due to the limitations of the device used (page 4, line 123 in the revised manuscript).
Figure 6
Comment 31: Please, indicate the histological staining and correct the magnification (20x and 40x instead of 200x and 400x)
Response 31: Thank you for your valuable feedback regarding the magnification details. The correct magnifications (20x) have now been provided, and the figure has been updated accordingly (page 7, lines 196-201 in the revised manuscript).
Reviewer 2 Report
Comments and Suggestions for Authors
In this case report, the authors have used for the first time in veterinary oncology, near-infrared fluorescence imaging (with indocyanine green) in a dog with gastric leiomyosarcoma, in order to improve in vivo tumor visualization and margin assessment, completed with positron emission tomography (PET), using 18F-fluorodeoxyglucose, in order to facilitate cancer staging. Two years after partial gastrectomy, the dog was alive, free of local recurrence and distant metastasis.
This manuscript is original, new in the field, very well written and illustrated. For instance, the authors have made considerable efforts to draw didactic representations of the tumor in its anatomical location (figure 2) and indocyanine green injection for near-infrared fluorescence imaging (figure 3) in complement to real pictures of surgical views and fluorescence imaging (figure 4). The discussion is very well written and informative.
Although there has been a problem in reference numbering in this manuscript, In my opinion this manuscript is fully in the scope of the special issue “Advances in Image-Guided Veterinary Surgery” edited by Animals, and of high quality.
Major comment:
1. Throughout the text, references are misnumbered. Examples: Page 1, line 40, introduction: reference 2 (Belia et al. 2022) does not apply to this sentence (about the relative frequency of gastric carcinomas in dogs). Page 1, line 41, introduction: reference 3 (Cassinotti et al. 2023) does not apply to this sentence (about late diagnosis of gastric carcinomas in dogs).
Minor comments/questions:
2. Page 3, line 100, case presentation: “or” instead of “of” (in the submucosa or the muscular layer).
3. Page 4, lines 142–143: “Local submucosal injections of warm saline were administered around the stomach mass”: can you please specify why this was done?
4. Page 4, lines 145–146: “The excised mass was immediately observed 145
5. under the NIR fluorescence imaging camera ex vivo”: can you please specify why this was done?
6. Page 5, line 150, title for figure 2: maybe replace “representative” by “schematic representation”.
7. Page 7, line 172, immunohistochemistry: please specify here the names of the clones used for CD117 (c-Kit) and DOG1 immunohistochemistry (if these antibodies are monoclonal), as well as the suppliers.
8. Page 7, lines 173–174, tumor diagnosis: please specify here that the tumor was not reactive to CD117 and DOG1. Theoretically, immunohistochemical detection of smooth muscle actin is necessary to obtain a definitive diagnosis of leiomyosarcoma. Page 7, line 175: please replace “per 10 high-power fields” by “in 2.37 mm2” (which is the standardized area chosen for mitotic counts in veterinary oncopathology). If the mitotic count was <1, I suppose it was 0 (nul)?
9. Page 7, line 180, legend for Figure 6A: on this picture, the tumor stroma is not dense at all. Line 180, legend for Figure 6B: this is not an IHC picture, but a closer view of the Hematoxylin-Eosin slide. Figures 6A and 6B: Please specify the length of the bar because it is not readable at all. Please have Figure 6 and its legend revised by your veterinary pathologist.
10. Page 7, line 182: I do not agree to use “chemotherapy” for. Toceranib / imatinib treatment: typically, this represents targeted therapy, not chemotherapy. Why “imatinib” instead of masitinib for a dog?
11. Page 7, line 183–184: the reasons given here (complete excision, absent lymphovascular invasion, no detected. Distant metastases) are OK for chemotherapy abstention. However, to exclude toceranib / masitinib treatment, the only valuable reason was that CD117 was not expressed by tumor cells.
12. Discussion: can you please discuss the availability of infrared fluorescence cameras for veterinary surgeons/oncologists?
Author Response
Major comment:
Comment 1: Throughout the text, references are misnumbered. Examples: Page 1, line 40, introduction: reference 2 (Belia et al. 2022) does not apply to this sentence (about the relative frequency of gastric carcinomas in dogs). Page 1, line 41, introduction: reference 3 (Cassinotti et al. 2023) does not apply to this sentence (about late diagnosis of gastric carcinomas in dogs).
Response 1: Thank you for your valuable feedback regarding the misnumbering of references. We have carefully reviewed and corrected the references throughout the manuscript to ensure their accuracy and relevance to the corresponding sentences.
Minor comments/questions:
Comment 2: Page 3, line 100, case presentation: “or” instead of “of” (in the submucosa or the muscular layer).
Response 2: Thank you for your suggestion. We revised this sentence as requested (page 3, line 106 in the revised manuscript).
Comment 3: Page 4, lines 142–143: “Local submucosal injections of warm saline were administered around the stomach mass”: can you please specify why this was done?
Response 3: We appreciate the reviewer’s comment. The local submucosal injection of warm saline around the stomach mass was performed to create a cushion, elevating the mass away from the deeper muscle layers. This ensures safer tissue manipulation, particularly in delicate areas like the stomach (Hirasawa, Ryuto, et al., 1997). We have incorporated this explanation and revised the manuscript accordingly (page 4, lines 154-156 in the revised manuscript).
Comment 4, 5: Page 4, lines 145–146: “The excised mass was immediately observed under the NIR fluorescence imaging camera ex vivo”: can you please specify why this was done?
Response 4, 5: We appreciate the reviewer’s insightful comment. The excised mass was immediately observed under the NIR fluorescence imaging camera ex vivo to confirm the presence of indocyanine green (ICG) fluorescence and ensure that the tumor margins were clearly identifiable. This step allows the surgeon to verify that the ICG has adequately highlighted the tumor area and can help confirm complete resection, minimizing the risk of residual tumor tissue. This clarification has been added to the manuscript (page 4, lines 157–161 in the revised manuscript).
Comment 6: Page 5, line 150, title for figure 2: maybe replace “representative” by “schematic representation”.
Response 6: We thank the reviewer for the suggestion. We have replaced "representative" with "schematic representation" in the title for Figure 2 as recommended (page 5, line 164 of the revised manuscript).
Comment 7: Page 7, line 172, immunohistochemistry: please specify here the names of the clones used for CD117 (c-Kit) and DOG1 immunohistochemistry (if these antibodies are monoclonal), as well as the suppliers.
Response 7: Thank you for your valuable comment. Upon inquiry with the laboratory, we confirmed the following information regarding the clones used for immunohistochemistry: CD117 (c-Kit): EP10; SMA: ASM-1; DOG-1: SP31. We have revised the manuscript to reflect these details (page 7, lines 188-190 in the revised manuscript).
Comment 8: Page 7, lines 173–174, tumor diagnosis: please specify here that the tumor was not reactive to CD117 and DOG1. Theoretically, immunohistochemical detection of smooth muscle actin is necessary to obtain a definitive diagnosis of leiomyosarcoma. Page 7, line 175: please replace “per 10 high-power fields” by “in 2.37 mm2” (which is the standardized area chosen for mitotic counts in veterinary oncopathology). If the mitotic count was <1, I suppose it was 0 (nul)?
Response 8:
We appreciate the reviewer’s insightful comment which improved the clarity and accuracy. In response, we have revised the manuscript to clarify that the tumor was immunonegative for CD117 (c-Kit) and DOG1 but displayed strong cytoplasmic immunolabeling for smooth muscle actin (aSMA), which is essential for the definitive diagnosis of leiomyosarcoma. Additionally, we have replaced the mitotic count description from “<1 per 10 high-power fields” to “0 in 2.37 mm²,” as per the standardized area for mitotic counts in veterinary oncopathology. These changes have been reflected in the revised manuscript (page 7, lines 190–195 in the revised manuscript).
Comment 9: Page 7, line 180, legend for Figure 6A: on this picture, the tumor stroma is not dense at all. Line 180, legend for Figure 6B: this is not an IHC picture, but a closer view of the Hematoxylin-Eosin slide. Figures 6A and 6B: Please specify the length of the bar because it is not readable at all. Please have Figure 6 and its legend revised by your veterinary pathologist.
Response 9: Thank you for your valuable feedback. Upon review, we have revised the figures and figure legends as suggested (page 7, lines 197-201 in the revised manuscript). We also attempted to address the bar length issue, but the original image files have been discarded. We reached out to our veterinary pathologist at IDEXX for assistance, and while new images can be obtained by cutting additional blocks and adjusting the magnification, this process will take over two weeks. Unfortunately, given the review submission timeline, we are unable to provide the updated images at this time. We appreciate your understanding and will make these adjustments in future publications if needed.
Comment 10: Page 7, line 182: I do not agree to use “chemotherapy” for. Toceranib / imatinib treatment: typically, this represents targeted therapy, not chemotherapy. Why “imatinib” instead of masitinib for a dog?
Response 10: We appreciate the reviewer’s comment and agree with the clarification. We have revised the terminology from “chemotherapy” to “targeted therapy” to accurately reflect the nature of the treatment (page 7, lines 202-203 in the revised manuscript). Additionally, while the option of targeted therapy was discussed, the management of such treatments is typically overseen by the oncology department, and therefore, the final decision regarding the specific therapeutic agents, including the choice between imatinib and masitinib, was made in consultation with oncologists.
Comment 11: Page 7, line 183–184: the reasons given here (complete excision, absent lymphovascular invasion, no detected. Distant metastases) are OK for chemotherapy abstention. However, to exclude toceranib / masitinib treatment, the only valuable reason was that CD117 was not expressed by tumor cells.
Response 11: We appreciate the reviewer’s valuable comment. We have revised the manuscript as suggested (Page 7, lines 202–205 in the revised version).
Comment 12: Discussion: can you please discuss the availability of infrared fluorescence cameras for veterinary surgeons/oncologists?
Response 12: We appreciate you for your insightful comments. We have revised the manuscript to incorporate a discussion on the availability of infrared fluorescence cameras for veterinary surgeons and oncologists, as suggested (page 9, lines 303-311 in the revised manuscript).
Round 2
Reviewer 1 Report
Comments and Suggestions for Authors
Many improvements have been made. In view of the previous comments, in my opinion, the clarity has been improved, the discussion has been strengthened, and all my concerns have been addressed. However, some minor revisions are required

Author Response
Comment 1: Lines 139-142: I suggest modifying the sentence as follows: “The dog was positioned in dorsal recumbency, and a celiotomy was performed from the xiphoid to the pubis, along with a left paracostal incision by extending through the external abdominal oblique muscle to access the cardiac region of the stomach. After palpating the mass at the esophagogastric junction, stay sutures were applied and a gastrotomy was performed to access the tumor, located in the mucosal layer of the stomach”.
Response 1: We appreciate your positive feedback and suggestions. We have revised the manuscript as suggested (page 4, lines 139-143 in the revised manuscript).
Comment 2: Line 145: please, replace “cc” with “ml”.
Response 2: Thank you for your suggestion. We have revised this as suggested (page 4, line 146 in the revised manuscript).
Comment 3: Lines 149-151: please, delete this sentence, as you have already described this in lines 142-143.
Response 3: We appreciate your suggestion. Thank you for your suggestion. We have deleted the sentence as requested.
Comment 4: Line 154: please, add “sterile” (Warm sterile saline).
Response 4: Thank you for your comment. We have revised this sentence as suggested (page 4, line 153 in the revised manuscript).
Comment 5: Line 159: you have already defined the abbreviation ICG, so it is not necessary to repeat it.
Response 5: We sincerely appreciate your attention to detail. As suggested, we have removed the repeated definition of the abbreviation "ICG" (page 4, line 158 in the revised manuscript).
Comment 6: Line 161: instead of PDS II, I would describe the type of suture used (3/0 absorbable monofilament), and then, in parentheses, I would include the brand of the suture (PDS II, Ethicon, USA). Additionally, the closure pattern of the abdominal wall, subcutaneous tissue, and skin should be specified.
Response 6: Thank you for your valuable suggestion. We have revised the sentence as suggested and included details regarding the suture type and closure pattern (page 4, lines 158-163 in the revised manuscript).
Comment 7: Line 244: I think “however” is not correct in this sentence. I should delate it and start the sentence with “The extended duration of the anaesthesia...”. Please, delete the “,” after “anesthesia”.
Response 7: We appreciate your suggestion and have revised the sentence accordingly (page 8, lines 246-248 in the revised manuscript).
Comment 8: Line 246: please, add a refence for “critical patients with underlying conditions such as renal or cardiovascular disease”.
Response 8: We appreciate your suggestion. We have added a reference to support the statement (page 8, lines 246-248 in the revised manuscript).
Comment 9: Line 302: please, add a reference after “minimize potential errors”.
Response 9: Thank you for your valuable feedback. We have added a reference as suggested (page 9, line 303 in the revised manuscript).
Comment 10: Lines 304-306: please, add a reference.
Response 10: We appreciate your suggestion regarding adding a reference for the importance of precision in the ICG injection technique to avoid complications. While we could not identify a specific reference that directly addresses this point, we believe it is essential to emphasize the critical nature of this technique in surgical practice. Therefore, we have decided to retain the content while removing the specific concerns about potential complications, such as bleeding and tumor cell spread, due to the absence of supporting references (page 9, lines 303-305 in the revised manuscript).
Comment 11: Line 322: as for line 159.
Response 11: Thank you for your comment. We have removed the repeated definition of the abbreviation "ICG" (page 9, line 323 in the revised manuscript).